# The Flt3L/Flt3 Axis in Dendritic Cell Biology and Cancer Immunotherapy

**DOI:** 10.3390/cancers13071525

**Published:** 2021-03-26

**Authors:** Francisco J. Cueto, David Sancho

**Affiliations:** Centro Nacional de Investigaciones Cardiovasculares (CNIC), 28029 Madrid, Spain

**Keywords:** Flt3, Flt3L, dendritic cells, cancer immunotherapy

## Abstract

**Simple Summary:**

Cancer immunotherapy is currently focused mainly on the enhancement of the effector function of T cells. However, dendritic cells (DCs) are needed to prime T cells, suggesting that DCs can be an attractive target for immunotherapy. Flt3L/Flt3 is an essential pathway for DC development and function, although its potential in cancer immunotherapy is not yet clearly established. Herein, we will review the current evidence which suggests that the stimulation of DCs through the Flt3/Flt3L axis may contribute to improved cancer immunotherapy.

**Abstract:**

Dendritic cells (DCs) prime anti-tumor T cell responses in tumor-draining lymph nodes and can restimulate T effector responses in the tumor site. Thus, in addition to unleashing T cell effector activity, current immunotherapies should be directed to boost DC function. Herein, we review the potential function of Flt3L as a tool for cancer immunotherapy. Flt3L is a growth factor that acts in Flt3-expressing multipotent progenitors and common lymphoid progenitors. Despite the broad expression of Flt3 in the hematopoietic progenitors, the main effect of the Flt3/Flt3L axis, revealed by the characterization of mice deficient in these genes, is the generation of conventional DCs (cDCs) and plasmacytoid DCs (pDCs). However, Flt3 signaling through PI3K and mTOR may also affect the function of mature DCs. We recapitulate the use of Flt3L in preclinical studies either as a single agent or in combination with other cancer therapies. We also analyze the use of Flt3L in clinical trials. The strong correlation between type 1 cDC (cDC1) infiltration of human cancers with overall survival in many cancer types suggests the potential use of Flt3L to boost expansion of this DC subset. However, this may need the combination of Flt3L with other immunomodulatory agents to boost cancer immunotherapy.

## 1. Introduction

The development and refinement of immunotherapy constitutes a revolution in the treatment of cancer. Most of the current immunotherapies target and unleash the effector function of lymphocytes. However, lymphocyte functions depend on their previous activation by antigen-presenting cells, among which dendritic cells (DCs) stand out. DCs were initially characterized according to their morphology and the expression of major histocompatibility complex class II and CD11c [1,2]. DCs continuously sample their microenvironment, where they can take up antigens and present them to T cells [3,4]. The outcome of the DC–T cell interaction can lead to immunity or tolerance depending on costimulatory signals present during the priming phase. Thus, DCs act as decision nodes in the initiation of T cell responses [3,4].

DCs derive from hematopoietic stem cells in the bone marrow (Figure 1) [5]. These can give rise to Lin^−^IL7Ra^−^Sca1^−^cKit^+^FcgR^lo^CD34^+^ common myeloid progenitors (CMPs), which can further differentiate into Lin^−^CX3CR1^+^CD11b^−^CSF1R^+^cKit^+^Flt3^+^ macrophage/DC progenitors (MDPs). MDPs can differentiate into Lin^−^CSF1R^+^cKit^lo^Flt3^+^ common DC progenitors (CDPs) which are completely committed to the DC lineage. CDPs can give rise to type 1 and type 2 conventional DCs (cDC1s and cDC2s) and plasmacytoid DCs (pDCs) [6].

cDC1 depends on basic leucine zipper ATF-like transcription factor 3 (BATF3) and interferon regulatory factor 8 (IRF8) and expresses X-C motif chemokine receptor 1 (XCR1) and dendritic cell natural killer lectin group receptor 1 (DNGR1; CLEC9A) in humans and mice, and BDCA3 (thrombomodulin; CD141) within the MHCII^+^CD11c^+^ population in humans. In mice, two cDC1 populations can be identified: a lymphoid tissue-resident population characterized by the expression of CD8α, and a peripheral population that expresses CD103. cDC1s stand out among other DC populations in their capacity to cross-present antigens and their production of IL12, which drives CD8 T cell responses, but are also key in CD4 T cell priming [7]. Batf3^−/−^ mice cannot generate peripheral cDC1s and the cell-associated cross-presentation by their resident cDC1s is impaired. These Batf3^−/−^ mice fail at controlling the development of highly immunogenic tumors, similar to Rag1^−/−^ mice that lack mature lymphocytes [8].

On the other hand, cDC2s constitute a group of cells that rely on V-rel reticuloendotheliosis viral oncogene homolog B (RELB), interferon regulatory factor 4 (IRF4), and zinc finger E-box-binding homeobox 2 (ZEB2) [9]. cDC2s express CD11b and signal regulatory protein alpha (SIRPα; CD172a) in mice, and BDCA1 (CD1c) in humans. Single-cell RNA sequencing technologies have enabled discrimination od a number of subsets within cDC2s [10,11]. These findings raise new questions about their differential developmental origin and their relevance in disease. Of note, a subset that expresses macrophage galactose N-acetyl-galactosamine specific lectin 2 (MGL2) displays outstanding antitumor capacities [11].

CDPs can also give rise to at least part of the pDCs, which can also derive from other lymphoid progenitors. pDCs are characterized by the expression of IL3 receptor alpha (IL3RA; CD123) and B220 in humans and mice, respectively. E2-2 and IRF8 are the main transcription factors involved in the development of pDCs. While pDCs are important producers of IFN-I during viral infections and IFN-I may have antitumor effects [12,13], cancers can disrupt the production of IFN-I by pDCs [14,15,16]. Furthermore, infiltration of pDCs within human tumors associates with poor prognosis [15,17].

The roles of the different DC subsets in orchestrating adaptive immune responses make them a very attractive target to boost antitumor immunity. Different strategies have been developed and evaluated in an effort to mobilize these populations, such as the administration of GM-CSF or the adoptive transfer of cells generated or stimulated in vitro. Among those efforts, we will focus this review on the use of Flt3L to enhance antitumor immunity.

## 2. Expression Pattern of Flt3 and Flt3L

The Fms-like tyrosine kinase receptor 3 (Flt3) was discovered as a surface protein intensely expressed on hematopoietic stem cells (HSCs) [18] (Figure 1). However, the improvements in the characterization of different hematopoietic progenitors identified CMPs and MDPs as the main expressors of Flt3 [19]. It should be noted that Flt3 expression is preserved in terminally differentiated cDCs and pDCs [20], which suggests that Flt3 signaling might have a functional impact on mature DCs. On the other hand, Flt3 is also expressed in some lymphoid progenitors. Although its function in this ontogeny branch has not been explored as profoundly, it plays a role in the early differentiation of B cells [21] and is re-expressed by B cells in the germinal center [22]. Moreover, new data indicate that the Flt3L/Flt3 axis is necessary for the development of NK and type 2 and 3 innate lymphoid cells [23] and their progenitors [24].

Flt3 belongs to the class III of tyrosine kinase receptors, characterized by a five immunoglobulin-domain extracellular region and a split tyrosine kinase domain. Class III tyrosine kinase receptors also include CSF1R, PDGFR, KIT and FMS [25,26]. Flt3 is encoded in chromosomes 13 in human and 5 in mice, encoding for 1000 and 993 aminoacid-long proteins, respectively [27].

Flt3^−/−^ mice show decreased numbers of pre-DCs (CD11c^int^CD45RA^lo^CD43^int^ SIRPα^int^CD4^−^CD8^−^MHCII^−^), cDCs, and pDCs in their spleens [19], as well as a clear drop in peripheral CD103^+^ cDC1s, but not CD11b^+^ cDC2s [20]. However, Flt3^−/−^ mice bear normal numbers of MDPs, indicating that Flt3 signaling is not required until the developmental stage of MDPs [19].

Two years after the discovery of Flt3, Lyman et al. identified a murine protein that could bind a soluble form of Flt3 [28]. This Flt3 ligand (Flt3L) promotes the expansion of Flt3^+^ primitive hematopoietic stem cells [28]. Soon afterwards, the human homolog of Flt3L (FLT3LG) was cloned, and a soluble version comprising its extracellular domain (amino acids 27 through 179) was found to induce proliferation in human CD34^+^ HSCs [29]. In the organism, Flt3L is expressed by multiple cell types, including stromal bone marrow and thymic cells [30,31], activated T lymphocytes [32], and NK cells [33], among others. Of note, blood levels of Flt3L are highly elevated in patients with aplastic anemia or receiving bone marrow-damaging chemotherapy or radiotherapy [34,35], and return to normal levels within three months from successful bone marrow transplantation [34].

In mice, three main isoforms have been reported [36]. Two of them contain a plasma membrane-spanning region and are tethered to the membrane [36]. The first one contains a cytoplasmic tail and its ectodomain can be cleaved to act as a soluble form [36]. The second membrane-bound isoform results from the retention of an intron during splicing, which limits the proteolytic release of the ectodomain [36]. TNFα-converting enzyme (TACE) mediates the shedding of the ectodomain of membrane-bound forms, which releases Flt3L ectodomain [37]. Accordingly, Tace^−/−^ mice display reduced levels of Flt3L in serum [37]. A third isoform lacks the membrane-spanning region and can be directly released from the cell [36]. In humans, the complete and the soluble isoforms have been identified, although not the one lacking the cytoplasmic domain [36].

Mice deficient in Flt3L show a dramatic absence of cDC1s, cDC2s and pDCs in both lymphoid and nonlymphoid tissues [20,38]. Flt3L^−/−^ mice also bear fewer CDPs and common lymphoid progenitors (CLPs, c-Kit^int^Flt3^+^CSF1R^lo^IL7Rα^+^) in their bone marrows [38]. Thus, the deficiency in the DC lineage in Flt3L^−/−^ mice is more profound than in Flt3^−/−^ mice, which has been attributed to a compensatory mechanism in Flt3^−/−^ mice, where DC progenitors become more sensitive to CSF1R signaling, which can compensate for Flt3 deficiency and promote DC development [39]. Under physiological conditions, NK cells, as well as other lymphocytes, have been identified as an important source of Flt3L within the tumor microenvironment (TME) [33]. The use of tumors that overexpress Flt3L has been extensively used to expand systemic cDC1s [40], but also tumor-infiltrating cDC1s [41]. Of note, several reports could not reproduce the expansion of cDC1s within the TME with systemic administration of Flt3L [42,43], which raises the possibility of some tumors becoming impervious to circulating Flt3L.

## 3. Flt3 Signaling in Response to Its Ligand

One of the main driving mutations in acute myeloid and acute lymphoid leukemias occurs in Flt3. This genetic alteration consists of internal tandem duplication sequences and is associated with poor prognosis in leukemia patients. Therefore, signaling pathways downstream of Flt3 have been extensively studied in the context of leukemia. However, physiological signaling through Flt3 in the context of DC generation remains largely unexplored. In the context of DCpoiesis, we will focus on homeostatic Flt3L signaling through Flt3.

As in most tyrosine kinase receptors, unstimulated Flt3 is thought to appear as a monomer on the plasma membrane, which renders its tyrosine kinase domain inactive [44]. Human Flt3L promotes the dimerization of Flt3 after binding through a compact binding domain fitting with the lock-and-key model [45,46]. Upon binding, Flt3L promotes the internalization of Flt3 receptors, which can be detected as soon as five minutes after the addition of Flt3L [46]. The internalized receptors are processed, and the products of their degradation can be observed around 20 min after the engagement of Flt3 and its ligand. The interaction of Flt3 with its ligand promotes the autophosphorylation of Flt3 tyrosine residues [47].

The first studies on the signaling cascade downstream of FLT3 used chimeric receptors composed of the extracellular domain of CSF1R and the transmembrane and cytoplasmic modules of FLT3 [48,49]. Upon the engagement of CSF1, the cytoplasmic domain of these chimeric receptors was found to bind phospholipase C gamma 1 (PLCγ1), the p85 subunit of phosphatidyl inositol 3′-kinase (PI3K), growth factor receptor-bound protein 2 (Grb2), and SHC1 [48,49]. This led to the phosphorylation of Ras GTPase-activating protein (GAP), Vav, Nck, Signal transducer and activator of transcription 5a (STAT5a), and SH2 domain-containing inositol phosphatase 1 (SHIP1), but no direct interaction with these proteins has been found [48,49,50,51]. Additionally, tyrosine residues in Gab1 and Gab2 become phosphorylated upon the FLT3/FLT3L engagement, which may act as adaptor proteins for Src homology region 2 domain-containing phosphatase 2 (SHP2), Grb2 and phosphatidylinositol 3 kinase (PI3K) [47]. Besides, Src homology 2 domain-containing transforming protein 1 (SHC) can interact and phosphorylate SHIP through its amino-terminal phosphotyrosine binding domain [50]. While most of the proteins identified downstream of Flt3 could potentially mediate its signaling, the phosphatases SHP2 and SHIP might act as negative regulators of the process.

Despite the great efforts directed to elucidating the signaling cascade triggered by Flt3, it has to be noted that most were performed in different cell lines that do not lead to the generation of mature DCs. The work in this context is more limited. Of note, Flt3L induces the phosphorylation of STAT3 during the generation of pDCs from bone marrow HSCs [52,53,54]. Accordingly, in mice, STAT3 is required for the development of pDCs, and its deficiency cannot be overcome by Flt3L administration [52].

Whether PI3K directly interacts or not with FLT3, the PI3K/mammalian target of the rapamycin (mTOR) cascade is clearly required for the generation of cDCs and pDCs by Flt3L [55]. In fact, CD11c-specific deletion of phosphatase and tensin homolog (PTEN), an Akt inhibitor that blocks the PI3K/mTOR pathway, expands the cDCs and pDCs [55]. In CD11c∆Pten mice, both lymphoid-resident CD8α^+^ and peripheral CD103^+^ cDC1s display a stronger expansion than CD11b^+^ cDC2s, which can be reverted by rapamycin [55].

## 4. Expansion of DCs with Flt3L

Flt3L fosters the expansion of granulocyte-macrophage colony-forming units (CFU-GM) and granulocyte, erythrocyte, monocyte, megakaryocyte colony-forming units (CFU-GEMM) [56], which ultimately reinforces the generation of DCs [57]. Especially, Flt3L was found to strongly expand a splenic DC population characterized by the co-expression of MHCII, CD11c, DEC205 and CD8α [57]. Flt3L can drive the expansion of various bone marrow progenitor populations, which results in the differentiation of B cells, NK cells, monocytes, red pulp macrophages, granulocytes, and innate lymphoid cells [23,38,56,58,59]. However, Flt3 expression is only conserved in mature cDCs and pDCs; not on B cells, monocytes, neutrophils, NK cells, or other innate lymphoid cells. In fact, the expansive effect of Flt3L has been reported to rely on the trans-presentation of IL15 by expanded DCs [60]. Focusing on the DC lineage, the administration of Flt3L promotes the expansion of MDPs in mice, which contributes to the expansion of both cDCs and pDCs [19]. Similarly, Flt3L can expand the amount of blood pre-DCs (CD11c^−^IL3Rα^+^ or SSC^lo^CD117^+^CD116^+^CD135^+^CD45RA^+^ CD115^−^) and CD141^+^ cDC1s, CD1c^+^ cDC2s and CD303^+^ pDCs in human volunteers, which can increase their frequencies even more than one order of magnitude [61,62].

The lymphoproliferative effects of Flt3L, together with the described mutations of the Flt3 receptor in different leukemias, suggest that its administration might promote lymphoproliferative malignancies. In mice, one report indicated this might be the case in mice inoculated with Flt3L-expressing retroviral vectors [63]. However, such effects have not been observed in the multiple studies where Flt3L was used for immunotherapeutic purposes. This might be caused by chronicity of the exposure to supraphysiologic levels of Flt3L and/or the use of irradiated mice and retroviral vectors, which can be tumorigenic on their own, or because of the local expression of Flt3L in the bone marrow. To the best of our knowledge, no clinical trial where Flt3L has been administered to patients or volunteers has reported the promotion of lymphoproliferative malignancies.

Addition of Flt3L to culture media drives the differentiation of bone marrow cell suspensions into the DC lineage [64]. Bone marrow cell cultures in the presence of Flt3L lead to mixtures of both cDC1s and cDC2s [64], with an important population of CD11c^+^B220^+^ pDCs that can produce IFN-I [65]. Flt3L culture-derived cDC1s resemble naturally occurring lymphoid tissue-resident cDC1s, but they lack the bona fide cDC1 markers CD8α and DEC205 [64,66]. Addition of Notch ligand Delta-like 1 to the standard Flt3L-supplemented media facilitates the generation of CD8α^+^DEC205^+^CD103^+^ cells, with capacity to migrate through CCR7 [67]. In addition, culturing CD34^+^ human cells with Flt3L required Notch signaling for full cDC1 generation, with granulocyte-macrophage colony-stimulating factor (GM-CSF) providing a synergistic effect [68]. Indeed, Flt3L can be used in combination with GM-CSF to generate CD103^+^ cDC1s from bone marrow cultures [69]. These CD103^+^ cDC1s are responsive to TLR stimulation, which drives the expression of activation markers such as CCR7, CD80 or CD86 [69].

## 5. Preclinical Studies Involving Flt3L

DCs are central to inducing T cell responses that might prevent cancer growth; therefore, Marakovsky et al. suggested that Flt3L might induce powerful antitumor immune responses [57]. Among the immune populations expanded by Flt3L, cDC1s stand out, whose infiltration within the TME has been extensively associated with patient survival. Flt3L did not affect the activation state of tumor-infiltrating cDC1s, because it did not affect the levels of CD40, CD86 or MHCII in tumor-infiltrating cDC1s [70]. However, it enhanced the proliferation of tumor-specific CD8 T cells at the tumor-draining lymph node [70], possibly by increasing the number of cross-presenting dendritic cells, especially cDC1s [41,71], although an effect of Flt3L on other immunomodulatory genes cannot be ruled out. Despite the frequent identification of cDC1s as the main cross-presenting DC population, other DC populations can promote antitumor CD8 T cell responses [41,71]. In fact, cDC1s express a cluster of BATF3-dependent genes, independent of IRF8-stabilization and cross-presentation, which are required for efficient antitumor immunity [72].

Indeed, administration of Flt3L as a single agent was demonstrated to delay or revert the growth of methylcholanthrene-induced fibrosarcomas [73], C3L5 breast tumors [74], B16 melanomas, and EL4 thymomas [75]. In these cases, the protection provided by Flt3L was associated with an expansion of DCs in both lymphoid and peripheral tissues, together with tumor antigen-specific T cell responses [73,74,75]. This observation has been recapitulated with tumor cell lines that stably express Flt3L, which regress after initial establishment [76]. However, some studies could not reproduce the protective effect of Flt3L as a single agent. An Flt3L-encoding adenovirus injected intravenously did not show any therapeutic effect on the cl-66 mammary tumor model [42]. Despite expanding DCs and other immune populations in spleen, this Flt3L-encoding adenovirus failed at promoting the infiltration of immune cells within tumors [42]. In this line, intraperitoneal administration of human recombinant Flt3L to mice failed at controlling the growth of CT26 and B16 tumors [43]. Once again, the administration of Flt3L did not increase the amount of MHCII^+^CD11c^+^ cells, which might include macrophages, within the TME, despite an intense increase in the number of circulating MHCII^+^CD11c^+^ DCs (up to 50-fold) [43]. In another study from the same laboratory, intraperitoneal administration of Flt3L enriched tumor-infiltrating cDC1s, but not cDC2s, within an ovalbumin-expressing B16 cell line [70]. The disparity of results obtained from preclinical models treated with Flt3L can be ascribed to differences in cancer models, Flt3L administration strategies, and dosage.

As with combination chemotherapy [77], a rational combination of immunotherapeutic strategies can synergize and provide increased protection against cancer [78]. In this case, it had been suggested that, by increasing the number of DCs, Flt3L might successfully synergize with other therapies that augment the availability of tumor antigens. Accordingly, Flt3L successfully increased survival after local radiation therapy in a metastasis model of Lewis lung carcinoma, which relied on antitumor T cell responses [79]. cDC1s are required for the abscopal (out-of-field) effect of radiation therapy, supporting that this might explain these benefits [80]. Additionally, recombinant human Flt3L has been administered to mice in combination with immunostimulatory DNA and tumor antigens to raise efficient antitumor immune responses that rely on CD8^+^ T cells and NK cells [81].

In the context of immune checkpoint blockades, tumor-infiltrating cDC1s have also been identified to play a pivotal role. Mice deficient in cDC1s do not respond to immunotherapy with anti-CTLA4 blocking antibodies [82]. Accordingly, in B16 melanomas and TRAMP prostate adenocarcinomas, anti-CTLA4 therapy can be improved by the inoculation of an Flt3L-expressing Vaccinia virus administered either intratumorally or subcutaneously in the opposite flank [83]. cDC1s are also required for other immune checkpoint therapies targeting PD1, PDL1 or 41BB [33,70,71]. cDC1-deficient mice displayed a reduced expression of PD1 in their tumor-infiltrating CD8 T cells [71], suggesting that cDC1s are required for a basal activation of cytotoxic immune responses. Besides cross-presentation, cDC1s excel at producing IL12 in the TME [41,84]. IL12 is necessary for immune checkpoint therapy [85], but not sufficient to restore responsiveness to immune checkpoint therapy in cDC1-deficient mice [71]. In several studies, a synergistic effect of Flt3L with polyI:C has been shown, whereby the cross-presentation of tumor antigens and tumor control is improved [70,71,86]. Here, a combination of Flt3L with TLR3 agonists improved the efficacy of anti-PDL1, anti-PD1 and anti-41BB [70,71,86]. However, in these reports, the precise contribution of Flt3L to the combination with immune checkpoint therapy is difficult to discern, because the antitumor effects of polyI:C do not rely on cDC1s [87]. Finally, Batf3-dependent cDC1s are also required for efficacious adoptive T cell transfer therapy [88,89]. Thus, adoptive transfer of Flt3L-secreting CD8 T cells expands tumor-infiltrating cDC1s and potentiates immunotherapy with polyI:C and 41BB-activating antibodies [76]. Overall, Flt3L constitutes a potential therapeutic agent for the treatment of cancer; multiple studies indicate that it improves antitumor immunity and restricts tumor growth. However, a better understanding of its mechanism of action might help prevent undesired interactions with other cancer therapies.

## 6. Flt3L in the Clinic

The potential therapeutic effect of Flt3L has given it access to several clinical trials (Table 1). In patients with metastatic colon cancer, pre-resection administration of Flt3L expanded both blood and perilesional DCs, but not tumor-infiltrating DCs [90]. In that study, Flt3L administration was associated with enhanced T cell immunity, shown by the increased sensitivity to recall antigens, but no objective response to Flt3L was observed [90]. Moreover, Fong et al. [91] reported that subcutaneous administration of recombinant Flt3L expands circulating DCs (HLADR^+^CD3^−^CD14^−^CD19^−^CD56^−^) in carcinoembryonic antigen (CEA)^+^ cancer patients. The increase in blood DCs for the generation of a leukapheresis product was loaded with CEA and reinfused into the patient [91]. Fong et al. allowed a two-day gap where they kept the DCs in culture to promote the upregulation of the costimulatory molecules CD80, CD86, CD40, CD83 and CMRF-44 and the chemokine receptor CCR7 [91]. Out of 12 patients, two showed tumor regression, one showed a mixed response, and in two the disease stabilized for at least three months, which associated with an enhanced cytotoxic potential from their peripheral blood mononuclear cells [91]. In another phase I clinical trial, Flt3L was administered for 14 consecutive days monthly to patients with human epidermal growth factor receptor 2 (HER-2/neu)^+^ breast and ovarian cancers [92]. Flt3L boosted the efficacy of a vaccine based on HER-2/neu peptides administered with GM-CSF [92].

Among its concerning adverse effects, Flt3L was suggested to drive lymphoproliferative malignancies [63]. Flt3L might promote the proliferation of Flt3L-dependent/-addict cancers, which is a frequent feature of acute myeloid leukemia. However, even some acute myeloid leukemia models can benefit from the immunostimulatory effects of Flt3L [93]. Thus, caution should be taken when administrating Flt3L to immune cell malignancies. Among the reported unwanted consequences, inflammatory effects stand out. Pre-resection administration of Flt3L to patients with metastatic colon cancers, only local symptoms such as erythematous nodules with no arthralgia, myalgia or fever were reported [90]. Fong et al. indicated that Flt3L administration only caused minor adverse effects, while DC reinfusion caused self-limited low-grade rigors and diarrhea [91]. In their study with gynecological cancer patients, Disis et al. [92] indicated that only one out of five patients receiving Flt3L alone developed a low-grade rash. Here, two out of five patients receiving Flt3L and GM-CSF developed serological alterations, with one of them suffering Sicca syndrome [92]. This study also reported the development of transient nonspecific autoimmune adverse effects in some patients [92]. As mentioned before, Bhardwaj and colleagues [94] tested the effect of Flt3L on patients receiving polyICLC, and described several low-grade local and systemic side effects, and even some high-grade effects such as anemia, hypophosphatemia, syncope, skin ulceration and sepsis. However, these are compatible with the administration of polyICLC [94]. Therefore, despite the concerns on the tumorigenic potential of Flt3L, all its adverse effects are associated with inflammatory disorders. However, all these reports agree that Flt3L was well tolerated, and no dose-limiting toxicity was observed.

In a recent study, in situ vaccination based on local administration of Flt3L, radiation therapy and the TLR3 ligand polyICLC showed increased CD8^+^ T cell responses in patients suffering indolent non-Hodgkin’s lymphoma [86]. This study reported systemic tumor regression in some patients, but not in those that had not responded to previous treatments [86]. These studies show strong evidence for a role for Flt3L in boosting antitumor immune responses, but it is important to note that trials which prove its clinical efficacy are still lacking.

## 7. Conclusions and Future Perspective

Due to the importance of DCs in antigen-(cross-)presenting populations, and the association of their infiltration within cancer with a positive prognosis, different compounds that drive the expansion of these populations has been actively pursued in the last years. To that aim, Flt3L is currently being evaluated in multiple clinical trials (e.g., NCT03789097, NCT02839265, NCT01976585) [86], but no clear therapeutic benefit has been reported.

Different results may rely on the administration route and dosage. In several reports, the administration of intravenous Flt3L did not show a therapeutic effect, but that might be due to the lack of DC expansion observed in those trials. It might be possible that a greater systemic dosage obtains an efficacious expansion of tumor-infiltrating cDC1s, while Flt3L expression in the TME, whether by tumor cells or adoptively transferred cells, might bring up the availability of Flt3L in situ.

In another line, Flt3L is a growth factor that is induced upon acute damage to the bone marrow. It has been shown that the seral levels of Flt3L are increased in patients with low blood cell counts, caused by aplastic anemia, chemotherapy, or radiation [34,35]. The success of bone marrow transplantation can also be tracked by determining the seral levels of Flt3L, which return to normal about three months after the procedure [34]. If chemotherapeutic drugs can drive the augmentation of Flt3L levels, the utility of exogenous Flt3L might be in the spotlight when combined with these therapies. However, it has been suggested that Flt3L can help overcome the toxicity of these therapies, similar to GM-CSF derivates.

Another aspect that remains unexplored is the effect of Flt3L on mature DC subsets, which maintain Flt3 expression. According to Cohen et al., STAT3 activation downstream of Flt3 is necessary for the efficient generation of cDC1s [96], although STAT3 signaling becomes a burden in differentiated cDC1s challenged with tumor-conditioned culture media [97]. Reviewing the significance of most of these studies is complicated, because they were carried out before a clear classification of DC had been established, and we have a better idea of which DC subsets are preferable to boost.

Another key factor that remains to be explored in depth is the impact of Flt3L in the expansion of tumor-infiltrating pDCs. Contrary to cDC1s, whose infiltration associates with good prognosis [41,82,84,98], and cDC2s, whose role in antitumor immunity is not clear [11], infiltration of human cancers by pDCs associates with poor prognosis [17]. Despite their potential as IFN-I producers, pDC functionality is impaired in the TME and instead drives Th2 and Treg immune responses through OX40L and ICOSL [15]. It is possible that Flt3L expands both cDC1s and pDCs with no net effect, but a rational combination of Flt3L with pDC-mediated immune checkpoints.

The relevance of DCs in driving antitumor immune responses, especially the strong association between cDC1 infiltration of human cancers with positive outcomes, maintain great expectation on the clinical utility of Flt3L. Despite the lack of positive results from clinical trials, designing rational therapeutic approaches through the combination of Flt3L with other immunomodulatory agents could have an impact in the war against cancer.

## Figures and Tables

**Figure 1 cancers-13-01525-f001:**
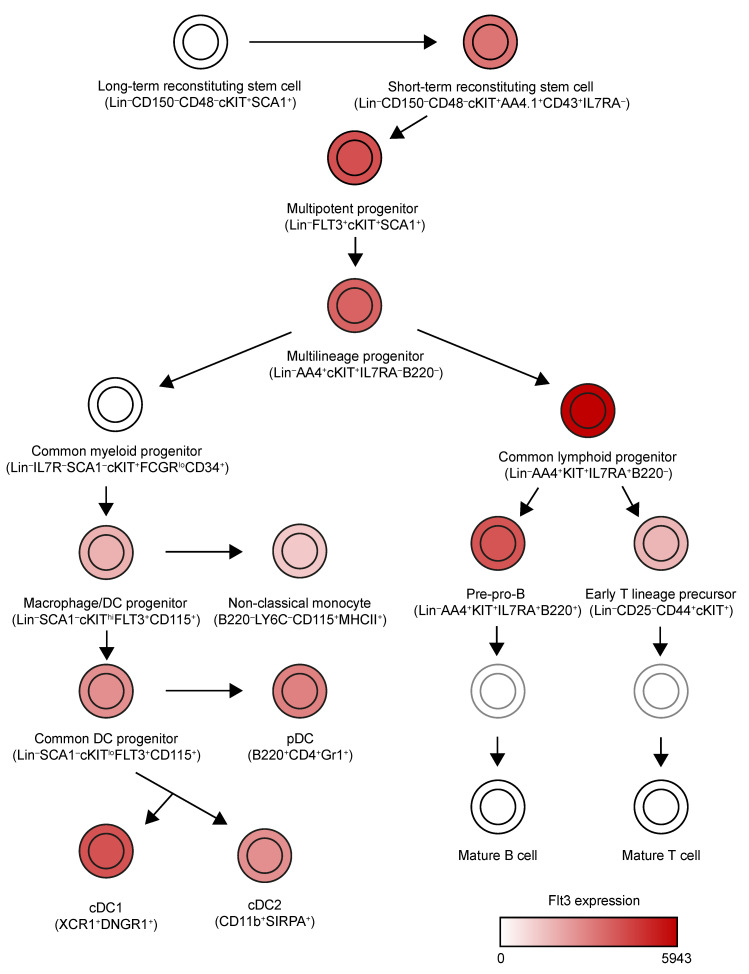
Expression pattern of Flt3 across mouse hematopoietic lineages. We have simplified the lineage tree generated by Jojic et al. [5]. This tree includes all cell populations identified by the first project of ImmGen to express high levels of *Flt3* (relative expression > 1000). In the case of mature cDC1s, cDC2s and pDCs, an average of Flt3 expression has been used to color them, because different populations from different organs were characterized in this consortium.

**Table 1 cancers-13-01525-t001:** Clinical Trials targeting the Flt3 receptor. Clinical trials registered in ClinicalTrials.gov where FLT3L has been used with immunostimulatory ends. This table includes clinical trials reported by 24 November 2020. Note that FLT3L is often used in a cytokine cocktail to expand hematopoietic precursors to improve the efficacy of bone marrow transplantation [95], but those have been excluded. s.c.: subcutaneous * Estimated accrual.

Identifier	Title	Indication	Therapeutic Strategy	Accrual	Clinical Trial Phase
**Status: Completed**					
NCT00006223	Flt3L in Treating Patients With Acute Myeloid Leukemia	Acute myeloid leukemia in remission	S.c. recombinant FLT3L vs. observation alone	139 *	III
NCT00003431	Flt3L in Treating Patients With Metastatic Colorectal Cancer	Metastatic colorectal cancer	S.c. recombinant FLT3L before resection of hepatic metastases	12 *	I
NCT00019396	Flt3L With or Without Vaccine Therapy in Treating Patients With Metastatic Melanoma or Renal Cell Cancer	Stage IV melanoma, stage IV renal cell cancer, recurrent renal cell cancer and recurrent melanoma	S.c. recombinant FLT3L alone or with melanoma-associated peptides	54–96 *	II
NCT00020540	Biological Therapy in Treating Patients With Metastatic Melanoma or Metastatic Kidney Cancer	Metastatic skin melanoma and metastatic kidney cancer	S.c. recombinant FLT3L with s.c. recombinant CD40L	5 *	I
NCT01465139	A Study to Evaluate CDX-301 (rhuFlt3L) in Healthy Volunteers	Healthy volunteers	Escalating doses of s.c. recombinant FLT3L (CDX-301)	30	I
NCT01484470	Umbilical Cord Transplantation for the Elderly Population	Multiple hematologic malignancies	Biological: StemEx	18	II
NCT02139267	Dose-finding, Safety Study of Plasmid DNA Therapeutic Vaccine to Treat Cervical Intraepithelial Neoplasia	Cervical intraepithelial neoplasia	Electroporation of DNA vaccine encoding for FLT3L and shuffled E6 and E7 genes of HPV type 16/18 (GX-188E)	72	II
**Status: Active, not Recruiting**					
NCT02839265	FLT3 Ligand Immunotherapy and Stereotactic Radiotherapy for Advanced Non-small Cell Lung Cancer	Advanced non-small cell lung cancer	S.c. recombinant FLT3L (CDX-301) with stereotactic body radiotherapy	29	II
NCT01811992	Combined Cytotoxic and Immune-Stimulatory Therapy for Glioma	Malignant glioma and glioblastoma multiforme	Dose escalation of adenovirus gene transfer that drives direct tumor killing and FLT3L expression	19	I
NCT02129075	CDX-1401 and Poly-ICLC Vaccine Therapy With or Without CDX-301 in Treating Patients With Stage IIB-IV Melanoma	Stage IIB-IV melanoma	S.c. recombinant FLT3L (CDX-301), s.c. or i.d. DEC-205/NY-ESO-1 fusion protein (CDX-1401) and s.c. poly-ICLC	60	II
**Status: Recruiting**					
NCT03789097	Vaccination With Flt3L, Radiation, and Poly-ICLC	Non-Hodgkin’s lymphoma, metastatic breast cancer and head and neck squamous cell carcinoma	In situ recombinant FLT3L, radiation and Poly ICLC with pembrolizumab	56 *	I/II
NCT01976585	In Situ Vaccine for Low-Grade Lymphoma: Combination of Intratumoral Flt3L and Poly-ICLC With Low-Dose Radiotherapy	Low-grade B-cell lymphoma	In situ recombinant FLT3L (CDX-301) and poly-ICLC	30 *	I/II
NCT03329950	A Study of CDX-1140 (CD40) as Monotherapy or in Combination in Patients With Advanced Malignancies	Multiple cancer types	CD40 agonist antibody (CDX-1140) alone vs. combination with recombinant FLT3L (CDX-301) vs. combination with pembrolizumab vs. combination with chemotherapy	260 *	I
**Status: not yet Recruiting**					
NCT04491084	FLT3 Ligand, CD40 Agonist Antibody, and Stereotactic Radiotherapy	Non-small cell lung cancer	FLT3L (CDX-301) with CD40 agonist antibody (CDX-1140) and stereotactic radiotherapy vs. stereotactic radiotherapy alone	46 *	I/II
NCT04616248	Radio-immunotherapy (CDX-301, Radiotherapy, CDX-1140 and Poly-ICLC) for the Treatment of Unresectable or Metastatic Breast Cancer Patients	Unresectable and metastatic breast cancer	In situ FLT3L, CD40 agonist antibody (CDX-1140), poly ICLC and radiation therapy vs. addition of i.v. CDX-1140	36 *	I
**Status: Unknown**					
NCT03206138	Safety and Efficacy of GX-188E Administered Via EP Plus GX-I7 or Imiquimod.	Cervical intraepithelial neoplasia 3	GX-188E with GX-I7 vs. GX-188E with imiquimod	50 *	
NCT02411019	Safety and Efficacy of GX-188E DNA Therapeutic Vaccine Administered by Electroporation After Observation	Cervical intraepithelial neoplasia 3	GX-188E	72	II

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
