# Peer review of "The Flt3L/Flt3 Axis in Dendritic Cell Biology and Cancer Immunotherapy"

_cancers, 2021, doi:10.3390/cancers13071525_

Round 1
Reviewer 1 Report
The authors extensively reviewed the expression, signaling, preclinical study and clinical trials of Flt3L in cancer therapies. The review is informative and provides significant contribution to the field. A minor suggestion is that the authors can use additional discussion regarding the interaction of Flt3+ DC cells with other immune cells, such as natural killer cells, in the tumor microenvironment.
Author Response
The authors extensively reviewed the expression, signaling, preclinical study and clinical trials of Flt3L in cancer therapies. The review is informative and provides significant contribution to the field.
Point 1: A minor suggestion is that the authors can use additional discussion regarding the interaction of Flt3+ DC cells with other immune cells, such as natural killer cells, in the tumor microenvironment.
Response: Thanks for the suggestion. We have included now one paragraph on the role of Flt3L on NK cell expansion and other immune populations in the TME (page 4, lines 136-142).
Reviewer 2 Report
The manuscript details the major features of DC cells and Flt3 signaling. The authors discussed the potential role of Flt3 in cancer therapy. Specifically, they focused their attention on immunotherapy.
I would develop this aspect more, as it is the focus of the paper.
It would be more correct to include figures and tables in the body of the manuscript.
Author Response
The manuscript details the major features of DC cells and Flt3 signaling. The authors discussed the potential role of Flt3 in cancer therapy. Specifically, they focused their attention on immunotherapy.
Point 2: I would develop this aspect more, as it is the focus of the paper.
Response: Thanks for the suggestions. We have now elaborated on this aspect and extended on the relevance of cDC1s in antitumor immunotherapies and how Flt3L can contribute to improve these (expanded in section 5, several paragraphs).
Point 3: It would be more correct to include figures and tables in the body of the manuscript.
Response: As requested, we have now moved Fig. 1 to page 3 and Table 1 to page 8.
Reviewer 3 Report
Regarding this review I have few comments for authors-
- Can author comment on if FLT3L can be used for immune cell malignancies?
- What is authors thought about that if FLT3L has been reported as a growth factor is it possible that for cancer cells it can also act as a growth factor and have adverse effect?
- As the FLT3L haven't showed clear clinical benefit and dosage increase can be a possibility. Is their any toxicity studies has been done for this ligand in clinic or can author comment on FLT3L related toxicity if there is any?
Author Response
Regarding this review I have few comments for authors-
Point 4: Can author comment on if FLT3L can be used for immune cell malignancies?
Response: Thanks for the suggestions. We have now included a commentary on that subject in pages 7 and 8 (lines 311-331)
Point 5: What is authors thought about that if FLT3L has been reported as a growth factor is it possible that for cancer cells it can also act as a growth factor and have adverse effect?
Response: We have included a commentary on that subject in pages 5 and 6 (lines 203-212).
Point 6: As the FLT3L haven't showed clear clinical benefit and dosage increase can be a possibility. Is their any toxicity studies has been done for this ligand in clinic or can author comment on FLT3L related toxicity if there is any?
Response: A new paragraph has been included to answer this important question (pages 7 and 8, lines 311-332).
Round 2
Reviewer 2 Report
All required points have been corrected or supplemented appropriately. The manuscript can be published